# Complementary performances of convolutional and capsule neural networks on classifying microfluidic images of dividing yeast cells

Mehran Ghafari[1]*, Justin Clark[1], Hao-Bo Guo[1], Ruofan Yu[2], Yu Sun[2], Weiwei Dang[2], Hong Qin[1,3]*

**1** SimCenter, Department of Computer Science and Engineering, University of Tennessee at Chattanooga, Chattanooga, Tennessee, United States of America, **2** Huffington Center on Aging, Baylor College of Medicine, Houston, Texas, United States of America, **3** Department of Computer Science and Engineering, Department of Biology, Geology and Environmental Science, University of Tennessee at Chattanooga, Chattanooga, Tennessee, United States of America

\* ryg668@mocs.utc.edu (MG); hong-qin@utc.edu (HQ)

**Data Availability Statement:** https://github.com/QinLab/GhafariClark2019.

**Funding:** This study was funded in part by the National Science Foundation in the form of grants

## Abstract

Microfluidic-based assays have become effective high-throughput approaches to examining replicative aging of budding yeast cells. Deep learning may offer an efficient way to analyze a large number of images collected from microfluidic experiments. Here, we compare three deep learning architectures to classify microfluidic time-lapse images of dividing yeast cells into categories that represent different stages in the yeast replicative aging process. We found that convolutional neural networks outperformed capsule networks in terms of accuracy, precision, and recall. The capsule networks had the most robust performance in detecting one specific category of cell images. An ensemble of three best-fitted single-architecture models achieves the highest overall accuracy, precision, and recall due to complementary performances. In addition, extending classification classes and data augmentation of the training dataset can improve the predictions of the biological categories in our study. This work lays a useful framework for sophisticated deep-learning processing of microfluidic-based assays of yeast replicative aging.

## Introduction

The budding yeast *Saccharomyces cerevisiae* is an effective model for studying cellular aging [1, 2]. The replicative lifespan of a yeast mother cell is defined as the total number of cell divisions accomplished or the number of daughter cells produced throughout its lifetime.

Microfluidics is a fast-developing technology for the single-cell monitoring and imaging required in this context. In particular, microfluidic devices are partially automatic method to monitor cells development and classify cells which can speed up the manual process of cells lifespan estimation [3].

(CAREER award #1453078, #1720215; and #1761839) awarded to HQ. This study was also funded by the National Institute of Health in the form of grants (Grant Nos. #R01AG052507 and #R42AG058368) awarded to WD. This study was also supported in the form of research support from University of Tennessee at Chattanooga awarded to HQ.

**Competing interests:** The authors have declared that no competing interests exist.

Typically, microfluidic images have relatively low resolution compared to confocal microscopic images that are often of high resolution [4], rendering unique challenges for microfluidics image processing [5]. For instance, microfluidic device materials, device coating, device volume, and area limitations increase capturing errors such as blurring, shifting focus, and trap deformation. Capturing the full progression of cellular replicative lifespans requires identifying both mother cells and daughter cells in full cell cycles [6]. Low image resolution hinders the automation of this process, demanding time-consuming, manual classifications of yeast replicative lifespans. Machine learning—specifically deep learning—could simplify this process.

Deep learning is a sub-field of machine learning that has been applied in a wide range of applications [7, 8], and its developments are mostly driven by computational capacity and the accessibility of datasets [9]. In recent years, deep learning has increased in efficacy for image classification and is now a popular method for parsing image information [10, 11]. Many innovations have been driven by creating models that perform well on benchmark datasets such as MNIST [12] (60,000 handwritten digits for training in a 28x28-dimensional vector space), CIFAR10 [13] (60,000 commonly used images in a 32x32-dimensional vector space), CIFAR100 [14] (500 training images grouped into 100 classes), ImageNet [15] (over 15M high-resolution images in over 22,000 classes), etc. The basic idea of deep learning is to create or "learn" a function that can map a high-dimensional input space into an output vector. For example, a high dimensional image can be filtered through neuron layers aiming for image classification and segmentation.

The Convolutional Neural Network (CNN) is one of the most frequent architectures used in image classification applications (e.g., medical images) where the output vector depends on the number of classes [16]. A variety of CNN approaches have proven useful for image classification because they are mainly designed for 2-dimensional (or higher) input tensors [17]. The proximity of pixels in the input images is also taken into consideration, which helps CNNs learn how pixels are oriented relative to each other, and leads to more accurate classification. One of the major drawbacks of CNNs is that they require a large number of training samples, a characteristic rooted in the architectural designs of CNNs [18]. The performance of a CNN model sometimes can be ameliorated by increasing the number of convolutional layers, which is computationally expensive. This requires some investigation and comparison between a CNN model with a low number of convolutional layers and a CNN model with a higher number of convolutional layers.

A different type of deep learning architecture, named CapsNet [19], was proposed to learn from fewer training samples than its traditional CNN counterparts. The recently proposed CapsNet architecture is known as capsule networks with dynamic routing. The model appears promising in image classification applications involving small datasets and still reaching a high level of accuracy [20]. The success of CapsNet lies in its ability to preserve additional information from input images by utilizing convolutional strides and dynamic routing instead of a max pooling layer. It has been argued that the spatial information of data has not been utilized in CNN models, including in the pooling function used to connect convolutional layers. For example, max pooling layers take only the most prominent values (e.g., pixels) from a previous convolutional kernel as input to the next layer. This issue considerably increases model inefficiency. In other words, CapsNet uses additional features of the dataset (e.g, spatial information) to improve the accuracy of a small dataset. These features are valuable especially in a medical dataset where there is a data limitation (e.g., images). CapsNet has illustrated improvement in accuracy on datasets such as MNIST, yet it is computationally expensive as training time increases substantially. In [19], the authors claimed that CapsNet can achieve near state-of-the-art performance on the MNIST dataset using 10% of the whole dataset.

The purpose of the current work is to compare deep-learning classification models of microfluidic images of dividing yeast cells. We compare three deep-learning neural network approaches to classify microfluidic trap images into 4 biological categories.

This comparative study focuses on the performance of three models: two convolutional neural networks and a capsule neural network. The two convolutional neural networks contains 2 and 13 convolutional layers respectively. We also investigated ensemble models built from these three models. Due to dataset limitations, we investigated the effect of data augmentation on all three models.

## Materials and methods

### Hardware and hyperparameters

All models were trained and tested on NVIDIA Tesla P100 GPU. We performed a basic grid search on six hyper-parameters: (1) the number of routing iterations, (2) learning rate, (3) batch size, (4) whether to add noise to training images, (5) the number of epochs in training, and (6) whether data augmentation was applied or not. The options of the hyper-parameter grid search are listed in S1 Table of the supporting information (SI). In general, a total of 108 combinations were initially tested.

### Dataset

The dataset is collected from a recent version of high-throughput yeast aging analysis (HYAA) chips experimental work [21]. Each time-lapse image has a resolution of 1280x960 and contains approximately 104 traps as shown in Fig 1A. In HYAA chips, traps are designed to hold a single dividing mother cell in direction of medium flow (top to bottom). The inlet width, outlet width, and height of each trap are 6, 3, and 5 micrometers, respectively. The outlet is wide enough to allow smaller daughter cells to slip through the trap outlet but narrow enough to withhold the bigger mother cell. Due to cell migrations (see S1A Fig), image intensity variations (see S1B Fig), low resolution, and difficulties in alignment, each time-lapse image is

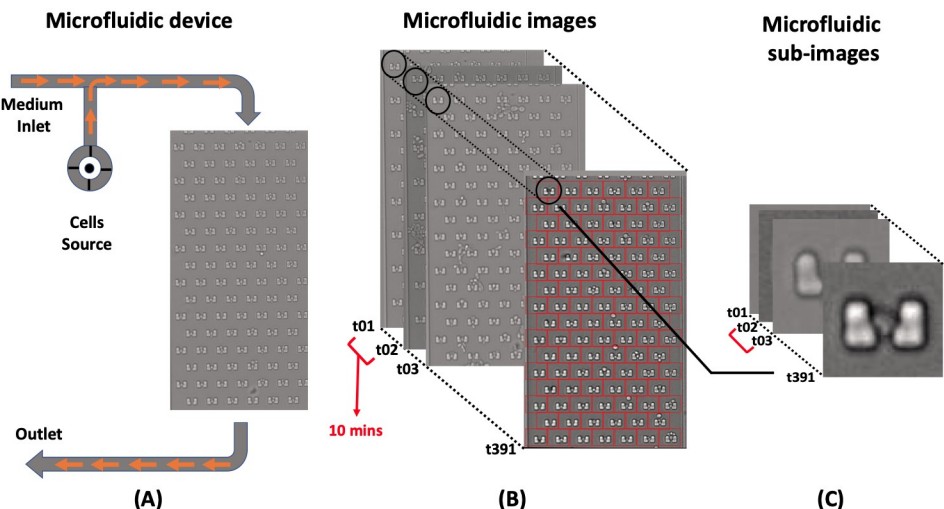

**Fig 1. The architecture of a microfluidics device.** (A) Single-channel microfluidic device with medium flow direction. Cells are inserted from cell source and joint medium before reaching the microfluidic traps. (B) Partitioning 104 traps of each microfluidic time-lapse images. (C) Time-lapse sub-images of a single trap in dimension of 60x60 pixels with 10-minute intervals.

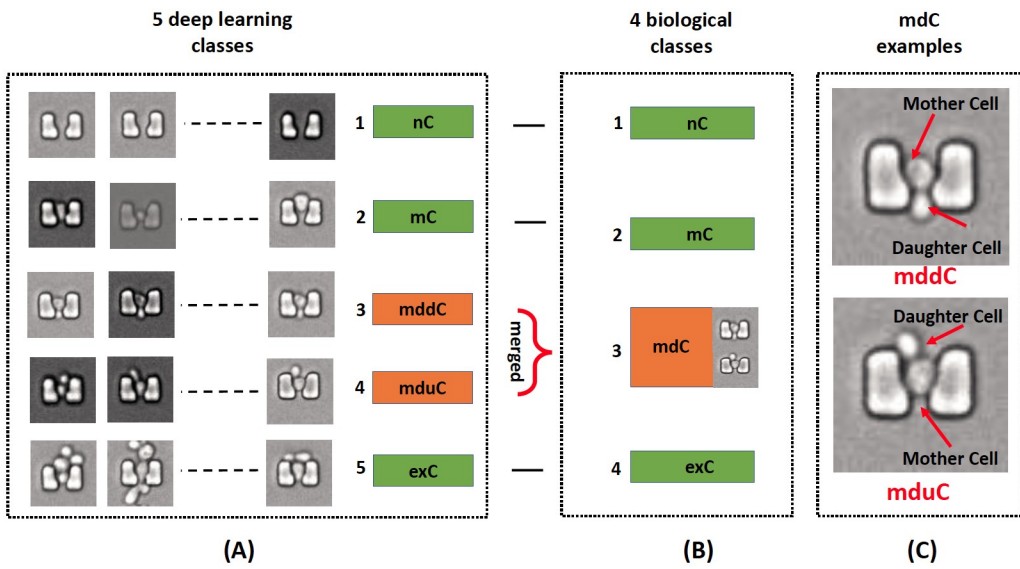

**Fig 2. Class categories with indication of results labeling for each class.** (A) 5 computed categories including nC, mC, mddC, mduC, and exC classes. (B) 4 biological categories including nC, mC, mdC, and exC classes. (C) An example of mddC and mduC for daughter cell orientation around a trap-center mother cell.

partitioned into sub-images of 60x60 pixels for an individual trap with respect to the boundary of its neighbor-traps as shown in Fig 1B. After partitioning, any individual trap typically contains 391 time-lapse sub-images with 10-minute intervals, which is illustrated in Fig 1C.

We trained the deep learning methods using 5 categories based on cell numbers and their relative positions: a trap with no cell (nC), a trap with a single mother cell (mC), a trap with one mother and one upward-oriented daughter cells (mduC), a trap with one mother and one downward-oriented daughter cells (mddC), and a trap with more than two cells (exC). We called all of these categories "5 deep learning classes," as illustrated in Fig 2A. The exC class is a holding category for any images that do not fall into nC, mC, mddC, or mduC. Although the mddC and mduC classes represent the same biological situation, their spatial patterns differ from each other such that separating these two situations leads to more consistent patterns when training deep learning models. Examples of mddC and mduC classes with an indication of cell positions are shown in Fig 2C. For biological purposes, when we constructed the confusion matrix, we merged the mddC and mduC classes that represent the same biological situation. Consequently, the confusion matrix is based on "4 biological classes" of nC, mC, mdC, and exC (Fig 2B).

## A 2-layered architecture, CNN-2

The two-layered architecture CNN that has two convolution layers represents one of the most simplified CNN models, and it is also referred to as the baseline CNN architecture [22, 23]. We chose this model for its simplicity, and we refer to it as the CNN-2 in the present work. The kernel size is 3x3 and batch normalization is applied to both layers [24, 25]. The strides for the first and second layers are 1 and 2 respectively, and the activation function is ReLU for this model. The input image size is 60x60 pixels and no image enhancement method is applied. A 2x2 kernel size used for max-pooling and 25% dropout applied for the second layer as the model architecture is shown in Fig 4A. We trained the model for 5, 10, and 20 epochs,

respectively; after 20 epochs there was no more improvement in accuracy and loss as shown in S2A and S2B Fig.

## A 13-layered architecture, CNN-13

We are aware of popular examples such as AlexNet [26], VGGNet [27], GoogleNet [28], etc. Each of these networks has tens to hundreds of millions of parameters (e.g., neural network weights) to learn and requires large training datasets. We chose a deep learning architecture termed the SimpleNet model [29], since it has additional 11 convolutional layers in comparison with CNN-2. HasanPour et al. [29] chose to think of the SimpleNet architecture in groups of layers, where each group of layers is homogeneous and thus can control overall network size and perform specific tasks well, such as classification and object detection. For clarity, we refer to SimpleNet as CNN-13 in our work. The CNN-13 architecture (see Fig 4B) is a convolutional neural network architecture with 13 layers. CNN-13 has 2–25 times fewer parameters than the popular models. We chose 2x2 and 3x3 kernels for pooling and convolutional layers respectively. The batch normalization and 25% dropout were applied to all layers. We trained the CNN-13 model for 5, 10, and 20 epochs, and after 20 epochs there was no more improvement in accuracy and loss as shown in S2C and S2D Fig.

## Capsule networks architecture

Capsule networks (CapsNet) is a novel architecture for deep learning. Basic versions of CapsNet have been shown to outperform extremely sophisticated CNN architectures [19]. A previous study showed that CapsNet could classify fluorescent microscopic images [30]. CapsNet replaces the typical pooling layer of CNNs with a more sophisticated weight-routing mechanism. As shown in Fig 3, instead of generating a scalar output as used in CNNs, a capsule layer in CapsNet generates a vector as output from convolutional kernel inputs.

The length of the vector represents the probability that a feature from the previous layer is present, and the values of the vector are an encoding of all the affine transformation of the kernel inputs. With a more data-efficient architecture (i.e., less information loss), fewer samples

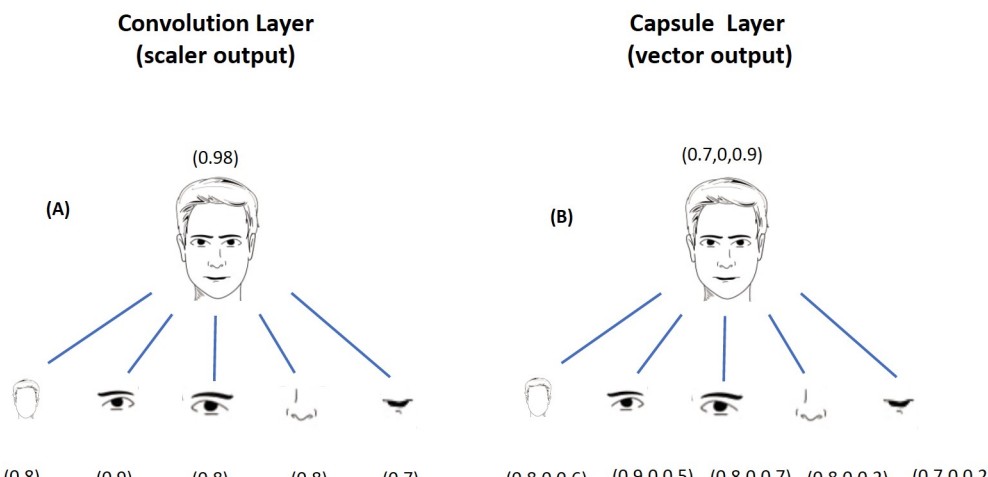

**Fig 3. CapsNet output comparison.** (A) The output of a CNN is a scalar. CNNs are transitionally invariant (shifting of an object does not affect output) and the learning becomes an enigmatic task when objects should be positioned relative to one another. (B) The output of a CapsNet is considered to be a vector. This renders additional information which the model can more easily learn the orientation of objects.

are required to train CapsNet models [30]. A non-linear "squashing" (Sj) function is used to minimize the length of the vector in the range of zero to one. The output vector (Vj) is calculated as:

$$\text{Vj} = \frac{||Sj||^2}{1 + ||Sj||^2} \frac{Sj}{||Sj||^2} \tag{1}$$

The squashing function can be calculated by

$$\text{Sj} = \sum_{i=1} Cij\hat{U}j|i \tag{2}$$

where Cij is a coupling coefficient and $\hat{U}j|i$ is a vector prediction for the output of the parent capsule. This vector can be calculated by

$$\hat{U}j|i = WijUi \tag{3}$$

where *Wij* and *Ui* are the weight matrix and the capsule output of the lower layer, respectively. We used the baseline CapsNet model as in previous works [19, 41] for our comparison studies. Fig 4C shows the architecture of the baseline CapsNet, which contains a convolution layer, primary capsule convolution and primary capsule reshape, DigitCaps (Squash function), and decoder. The kernel size is 9x9 and the stride is 2 for primary capsule convolution. The dimension for primary capsule reshape is 22x22x32 with 8 capsules. A grid search of the hyper-parameters (see S1 Table) led to 108 trained CapsNet models, from which we picked 10 top-

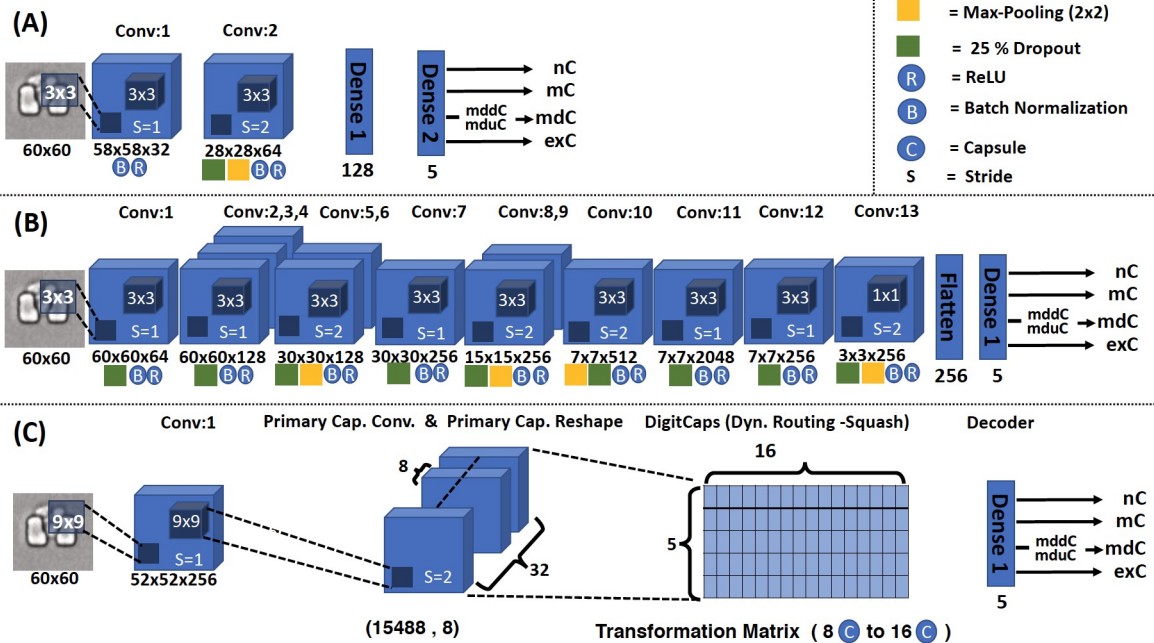

**Fig 4. Architectures of three models.** (A) CNN-2: A total of 2 convolutional layers and 2 densely connected layers. (B) CNN-13: A total of 13 convolutional layers plus a densely connected layer. (C) Capsule Network: A convolutional layer plus a high-level capsule layer and a densely connected layer. In general, CapsNet contains two parts: the encoder that takes an input image and learns to encode it into 16D instantiating vector parameters, and the decoder that takes a correct DigitCap from a 16D vector and learns to decode it into an original-like image.

performing models. We then examined these 10 models and picked the best-performing Caps-Net model for further studies.

## Data augmentation

Due to the tedious process of manual annotation, we have a relatively small number of training images. Several affine transformations applied to augment training images [31, 32]. Affine transformations on the original images are a popular and simple data augmentation method [33]. The data augmentation table for this work is available in S1 Table. In general, noise added to images and applied feature center, Std normalization, rotation, width shift, height shift, brightness, horizontal flip, and vertical flip on the training images. The total number of trap images in our datasets is 1,000 for each of the five categories. We used 3,078 images for training (1,026 images for validation), and 896 images for testing. The training data augmentation resulted in 99,380 training images. The codes and dataset of this work are available from https://github.com/QinLab/GhafariClark2019.

## Performance metrics

Three key metrics have been used in the model analysis [34]. The first is accuracy, e.g., the number of true positive and true negative exC predictions versus all of the exC examples. The second metric is precision, e.g., the true positives prediction of the mC class versus all true positives and false positives of mC. Lastly, we are concerned with a metric called recall [35]. One example of recall is the true positives prediction of the mdC class versus all true positives and false negatives of mdC. Each of these three metrics has its purpose, and they are oftentimes used together to determine the overall performance of a model [36], written as

$$\text{Accuracy} = \frac{TP + TN}{TP + TN + FP + FN}, \tag{4}$$

$$\text{Precision} = \frac{TP}{TP + FP}, \tag{5}$$

$$\text{Recall} = \frac{TP}{TP + FN}, \tag{6}$$

where TP, TN, FP, and FN refer to true positives, true negatives, false positives, and false negatives respectively. Moreover, F1 can simply be calculated from Eqs 5 and 6.

$$\text{F1} = 2\frac{(Precision)(Recall)}{Precision + Recall} \tag{7}$$

In this work, F1 values can be calculated from the result of precision and recall using Eq 7.

## Results and discussion

### Extension of classes improved the accuracy of predicted biological categories

At the initial stage, all models were trained and tested with 4 deep learning classes: nC, mC, mdC, and exC. Here, mdC refers to any traps with two cells without merging any classes, as in Fig 2C where all orientation of daughter cell around the mother cell (inside or outside trap) is considered as class mdC. However, early in the process of model selection and tuning, many training images were misclassified when two cells were observed inside the same trap. Some of

the best models struggled to reach 60% test accuracy. One approach to reducing misclassification is to use transfer learning [37]. The other concept is splitting classes and using pre-trained weight. This method has some similarity to transfer learning as both methods attempt to make it easy for the models to learn weights; however, these approaches come from different angles. For instance, there are size, pattern, and orientation similarities between the exC class and the mdC cell class. In many cases, a single mother cell appears as two cells (due to dynamic shape and low image resolution) when the daughter cell is above or below the mother cell. Based on these observations, we split all images with two cells into two separate classes; in the first class, the daughter cells are on top of mother cells (upward-oriented, mduC class), and in the second class the daughter cells are below the mother cells (downward-oriented, mddC class), as illustrated in Fig 2C. At the highest level, creating mddC and mduC classes improved the homogeneity of the two classes and helped the situation where the neural networks were able to more easily learn the differences of the mduC class and the exC class without having to learn that the mddC and mduC class are the same.

It is important to notice that all training and testing activities are based on the computed 5 classes dataset. Since there is no biological difference between mddC and mduC classes, the results for mddC and mduC classes are merged and labeled as mdC for easier biological understanding as shown in Fig 2B.

## CNN-2 performance was improved by training data augmentation

CNN-2 exhibited instability and did not perform well when it was trained with non-augmented training datasets as shown in S2A and S2B Fig. Fig 5 represents the overall performance of the CNN-2 model without data augmentation in green bars. Initially, CNN-2 model trained without data augmentation performed poorly in the mC, with precision at 71% and recall at 66%. The comparison results in Fig 6 indicate that data augmentation mainly improved the accuracy of prediction over the mdC class in this model. As a result of the training data augmentation, the overall accuracy of CNN-2 was improved from 87% to 90.29% (Fig 6D). The misclassification results show that two common types of misclassification occurred in CNN-2 while there were only two cells observed inside the trap. For S5 Table CNN-2 (A), the model wrongly predicted two cells instead of three cells due to blurred boundaries. Cases in S5 Table CNN-2 (B) and S5 Table (C) were a little more problematic because the CNN-2 model did not recognize the daughter cells above or below the mother cells. Interestingly, for S5 Table CNN-2 (D), the mother cell is almost entirely transparent and ends up not being a problem after recombining the mddC and mduC classes.

## CNN-13 performance and impact of training data augmentation

CNN-13 showed substantial improvement in average accuracy in comparison to CNN-2, and this improvement occurred for CNN-13 models trained with and without augmentation of training datasets, as shown in Figs 5 and 6. Augmentation of the training dataset also led to more stable CNN-13 models as seen when changes of the cost functions during training became more smooth with augmented datasets as shown in S2C and S2D Fig. Surprisingly, data augmentation had a marginal effect on the accuracy, precision, and recall of CNN-13 Fig 6. Without data augmentation, the model predicted 100% in nC class (precision and recall) and exC class (precision). Most of the misclassification appears to be in the mC and mdC classes. With data augmentation, prediction for the nC did not change (100%) and the mC recall improved from 93% to 96% (precision had the opposite reaction). Furthermore, S2 Table shows that data augmentation had a slight improvement in the mC and exC classes but a

# Test set results ( without data augmentation)

**Fig 5. Test set results of models classification without data augmentation based on 4 biological classes.** (A) The table represents the correct and misprediction results of the mddC and mduC classes without data augmentation for all three models. The orange color indicates the predicted mduC class, and the blue color indicates the predicted mddC class based on correct prediction and misprediction. (B) The bar graph represents the highest precision for the exC class and lowest prediction for the mdC class based on 4 biological classes. (C) The bar graph represents the highest recall for the nC class and lowest recall for the mC class based on 4 biological classes. (D) It shows that the overall precision, recall, and accuracy for CNN-13 are higher than the other two models. (E) It illustrates the total number of test set images, total predicted images, and total mispredicted images for each model.

negative effect in the mdC class. The overall accuracy for this model was 97% without data augmentation and 98% with data augmentation as shown in purple bars (Fig 6).

Considering misclassification for CNN-13, S5 Table CNN-13 (A) shows several cells clustered together. After further inspection, this image was classified with near 100% certainty. Although this instance is uncommon, it still poses problems in cell type identification. The mistake on S5 Table CNN-13 (B) is more understandable since there is a mother cell with seemingly two daughter cells on top. The algorithm did not classify this example in the exC class and instead predicted it as the mduC. Since one of these cells could be a true daughter cell, this image may not be as problematic. Image S3 Table CNN-13 (C) is similar to the previous image, but the boundary between the two cells on top of the mother cell are so thin that it is reasonable to think that it is a deformed single daughter cell to the untrained eye. Lastly, S5 Table CNN-13 (D) illustrates a mistake that was common in the CNN-2 model where the mduC or mddC were predicted as the mC due to blurred boundaries.

## CapsNet performance and impact of training data augmentation

The performance of CapsNet was more sensitive to hyper-parameters than were the CNN-2 and CNN-13 models, based on grid searches on the hyper-parameters detailed in S1 Table. We picked the best-performing CapsNet model for this study. The training data augmentation mainly improved CapsNet accuracy of the mC category but not of other categories (Fig 6). The overall accuracy of CapsNet reached 90% with data augmentation. In Zhang et al. [38], a close

# Test set results (with data augmentation)

**Fig 6. The test set results of classification models with data augmentation based on 4 biological classes.** (A) The table presents the correct and misprediction results of the mddC and mduC classes with data augmentation for all three models. The orange color indicates the predicted mduC class, and the blue color indicates the predicted mddC class based on correct prediction and misprediction. (B) The bar graph represents the highest precision for exC class and the lowest prediction for the mdC class with a similar ratio without data augmentation. (C) The bar graph represents the highest recall for the nC class and the lowest recall for the mC class. (D) It shows that the overall precision, recall, and accuracy for the ensemble model are higher than the other three models. (E) It illustrates the total number of test set images, total predicted images, and total mispredicted images for each model with data augmentation. The misprediction results for CNN-2, CNN-13, and CapsNet were one of the motivations to generate ensemble models. In the bar graphs, the precision and recall show noticeable improvement for all models after data augmentation. Every single well-performing model had an augmented dataset. Overall, the data augmentation mainly improved CNN-2 and CapsNet models.

range of accuracy was reported on fluorescent images with a different number of images for training and test sets.

In one case of misclassification, S5 Table CapsNet (A) shows that there is a small cell on the top right portion of the mother cell that seemed to be overlooked by the CapsNet model. One potential cause for this misclassification is that the two cells on top of the mother cell are quite different in size. S5 Table CapsNet (B) is one of the problematic misclassifications that CNN-13 was good at detecting. S5 Table CapsNet (C) shows a transparent cell that could be a senescent cell or dead cell. This type of image is unlikely to happen often enough for the model to learn effectively. S5 Table CapsNet (D) shows another interesting example. The oversized mother cell that appears almost at the outlet of the trap is reproducing a daughter that flows over the outside edge of the trap, which increases the probability of misclassification.

## Deeper layers bring moderate improvement and challenging performance of the CapsNet

S4 Table presents the test accuracy results without and with an augmented training dataset for individual biological classes. The table shows that the CNN-13 performed well and most of the predictions are above 92% for all classes without and with data augmentation. The CNN-2 and CapsNet had a weaker performance, as the accuracy for one of the classes is below 70% (e.g., mC). In contrast, CNN-2 can predict the nC category with 100% accuracy (see S3 Table). The

performance of CNN-2 can be greatly improved by data augmentation and adding convolutional layers. As expected by the increased number of convolutional layers, CNN-13 had greater overall accuracy than CNN-2, as shown by its confusion matrix (see S3 Table). With the additional 11 convolutional layers and much more training time, CNN-13 improved the overall accuracy to 98%, a partial 6% increase from CNN-2 for an additional computational cost (S2 Fig). The performance of CNN-13 is not substantially changed by applying data augmentation. Fig 6 shows that data augmentation improved the total prediction of CNN-13 by 0.22%, which is around 16 times lower than CNN-2, and decreased the total misprediction by 8.3%, which is considerably lower than CNN-2. On the other hand, CapsNet was the weakest model in terms of average accuracy, and training time was twice of CNN-13. According to the confusion matrix (see S3 Table), the model only had a great prediction for the nC (180/180). Surprisingly, the model had the best prediction (354/360) for the mdC class without data augmentation where both CNN-2 and CNN-13 struggled with the prediction (with or without data augmentation). Still, CapsNet had poor prediction for the mC and exC classes. Fig 6 illustrates that the data augmentation was an effective approach that improved the total prediction by 7% (better than the CNN-2 model) and decreased the total misprediction by 30.8% (better than the other two CNN models). CapsNet is much more sensitive to data augmentation than the other two CNN models are, and it can perform well on a specific class.

## Each deep learning model has its own profiles of misclassifications

We also investigated the misclassification behavior of individual models for the mC, mdC, and exC classes as illustrated in Figs 5 and 6. In terms of correct-prediction balance between the mddC and mduC, Figs 5A and 6A demonstrate that all the models had relatively close range of prediction for the mddC and mduC (without and with data augmentation). In terms of misprediction, the CNN-2 model had the opposite behavior of the CNN-13 and CapsNet models. For CNN-2, the mduC class had a higher percentage of misclassification for the mC class, and the exC class had higher misclassification for the mddC class. For CNN-13 and CapsNet, the mddC class had a higher percentage of misclassification for the mC class, and the exC class had a higher misclassification for the mduC. These comparisons indicate that why we consider an ensemble model as an alternative.

## Ensemble models performance

In machine learning, minimizing bias and variance errors is a challenging task. The weighted average ensemble model is one of the methods to overcome this issue that relies on two properties in machine learning [39]: creating an ensemble model such that the bias can be decreased at expense of increased variance, and creating an ensemble model such that the variance can be decreased at no expense to bias [40]. In general, there are two simple methods to combine several machine learning models and create an ensemble model with better performance. First, train a model (e.g., classifier) over multiple subsets of the training dataset, which leads to different models. Then, the individual model can have a prediction on the test dataset and the results can be averaged as an ensemble model. This method is useful when there is no other model available. The other method is to train various models on the same dataset and average the results on the test dataset. An ensemble model attains a synergistic betterment in overall performance including reproducibility and stability.

Inasmuch as each single deep learning model had an uneven performance in the 4 biological classes, we considered investigating the combination models to achieve greater performance. There are four different possibilities to combine these three single deep learning models (see S3 Fig). We tested the combinations with data augmentation since it improved the

performance of the individual models. The results from all three models indicate that weighting the predictions by overall model accuracy achieves slightly better performance [41]. Thus, models in the ensembles presented are weighted by their overall validation set accuracy and applied to the test set. The CNN-13 predictions had the highest weight, the CNN-2 was weighted slightly lower, and CapsNet had the lowest prediction weights. Therefore, the three-member ensemble, No.4 (see S3 Fig), outperformed all of the two-member ensembles. Since the result of ensembles 1 to 3 were almost similar to the individual models, we only represent the result of ensemble No. 4 here. The ensemble model results in yellow color (bar graphs) from Fig 6 show that the overall accuracy of ensemble No.4 is 98.5% (better than the CNN-13). The precision result is better than other models for the nC, mdC, and exC classes except for the mC class in comparison to the CNN-13 model. Similarly, the model had greater recall results for all biological classes. In terms of ensemble No.4 misclassification, each of the ensemble models has misclassification of at least one of the three models.

## Future work

While correctly classifying images into one of the four discussed categories was the focus of this work, there are still improvements to be made in image pre-processing (e.g., image resolution). In addition, we could improve the overall ensemble by adding more diversity to the set of models. For example, the sequential nature of the problem could lend itself nicely to a Long Short-Term Memory (LSTM) [42] and convolutional LSTM architectures.

## Conclusion

We compared three deep learning models for the classification of microfluidic images of dividing yeast cells. Microfluidic images are typically low resolution, which poses challenges for computational analysis. We discovered that data augmentation of training data can improve the performance of both convolutional and capsule networks. In addition, splitting a class into two classes could be an effective approach for some models based on the type of dataset and model architecture. We evaluated that a baseline architecture of a convolutional network with two layers could give 90+% overall accuracy and deep layered convolutional networks could improve the overall accuracy at the expense of substantially more computing cost. Moreover, the baseline architecture of capsule neural networks did not outperform the deep-layered convolutional networks in terms of overall accuracy, though the baseline capsule networks could detect a specific type of data with better performance. Consequently, an ensemble model reached 98.5% overall accuracy by combining the strengths of different models. We showed that an ensemble of the top three models performs better than using each model alone, leading to a good "collaboration" among these models. Overall, convolutional and capsule neural networks have complementary performances for the classification of microfluidic images of dividing yeast cells.

## Supporting information

**S1 Fig. Microfluidic images.** (A) Time-lapsed images from time-point 001 to time-point 391. Black circles with connected dash-lines indicate that some of the traps become overcrowded over time. (B) Each image partitioned to 60x60 pixels sub-images, and individual trap image is highly variable. While traps and cells have a limited number of orientations, the contrast, brightness, and image quality all add great complexity to the dataset. There are often shadows, depending on the lighting conditions of the experiment.
(TIF)

**S2 Fig. CNN-2 and CNN-13 training and test plots.** (A) and (B) are plots for the CNN-2 model without and with data augmentation. (C) and (D) are plots for CNN-13 model without and with data augmentation.
(TIF)

**S3 Fig. Ensemble models combination.** Results of CNN-2, CNN-13, and CapsNet models indicated that there are numerous ways to ensemble (i.e., combine) models together to create a single aggregate model. We explored the results from all possible ensembles with different combinations based on practical and key performance metrics.
(TIF)

**S1 Table. Grid search and data augmentation options.** The grid search option table used for all models and data augmentation features applied when the data augmentation in grid search option was set to "True."
(TIF)

**S2 Table. All models comparison.** The results of each model for individual biological class with effect of data augmentation.
(TIF)

**S3 Table. Models confusion matrix.** Three models confusion matrix with indication of data augmentation effectiveness.
(TIF)

**S4 Table. Models accuracy.** Accuracy of models for individual biological class.
(TIF)

**S5 Table. Sample image of most common misclassifications.** CNN-2 (A) label exC: prediction mduC, CNN-2 (B) label mduC: prediction mC, CNN-2 (C) label mddC: prediction mC, CNN-2 (D) label mduC: predicted mddC. CNN-13 (A) label exC: prediction mduC, CNN-13 (B) label exC: prediction mduC, CNN-13 (C) label exC: prediction mduC, CNN-13 (D) label mddC: prediction mC. CapsNet (A) label exC: prediction mduC, CapsNet (B) label mddC: prediction mC, CapsNet (C) label mC: prediction mduC, CapsNet (D) label mduC: prediction exC. Ensemble No.4 (A) label mdC: prediction mC, Ensemble No.4 (B) label exC: prediction mdC, Ensemble No.4 (C) label exC: prediction mdC, Ensemble No.4 (D) label mdC: prediction exC.
(TIF)

## Acknowledgments

The authors would like to thank the computing facility of the SimCenter at the University of Tennessee at Chattanooga. We also thank Bailey S. Kirby for her editorial support.

## Author Contributions

**Conceptualization:** Mehran Ghafari, Justin Clark, Hong Qin.

**Data curation:** Ruofan Yu, Yu Sun, Weiwei Dang, Hong Qin.

**Formal analysis:** Mehran Ghafari, Hong Qin.

**Funding acquisition:** Mehran Ghafari.

**Investigation:** Mehran Ghafari, Justin Clark.

**Methodology:** Mehran Ghafari.

**Project administration:** Hong Qin.

**Software:** Mehran Ghafari, Justin Clark.

**Supervision:** Hong Qin.

**Visualization:** Mehran Ghafari.

**Writing – original draft:** Mehran Ghafari, Justin Clark.

**Writing – review & editing:** Mehran Ghafari, Hao-Bo Guo, Hong Qin.

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
