## [Decision Letter · Decision Letter 0]

24 Nov 2020

PONE-D-20-27932

Complementary Performances of Convolutional and Capsule Neural Networks on Classifying Microfluidic Images of Dividing Yeast Cells

PLOS ONE

Dear Dr. Ghafari,

Thank you for submitting your manuscript to PLOS ONE. After careful consideration, we feel that it has merit but does not fully meet PLOS ONE’s publication criteria as it currently stands. Therefore, we invite you to submit a revised version of the manuscript that addresses the points raised during the review process.

We look forward to receiving your revised manuscript.

Kind regards,

Friedhelm Schwenker

Academic Editor

PLOS ONE

Journal Requirements:

Reviewers' comments:

Reviewer's Responses to Questions

**Comments to the Author**

1. Is the manuscript technically sound, and do the data support the conclusions?

Reviewer #1: Yes

Reviewer #2: Yes

2. Has the statistical analysis been performed appropriately and rigorously? 

Reviewer #1: Yes

Reviewer #2: Yes

3. Have the authors made all data underlying the findings in their manuscript fully available?

Reviewer #1: Yes

Reviewer #2: Yes

4. Is the manuscript presented in an intelligible fashion and written in standard English?

Reviewer #1: Yes

Reviewer #2: No

5. Review Comments to the Author

Reviewer #1: 1 The CapsNet (2017) is not the latest and the most effective. This domain develops fast and the authors should keep an eye on the literature.

2 The methods of other people are not sufficient as highlights of study.

3 It is better that the authors can excavate certain medical meaning of this study.

4 The ensemble method proposed by the authors, is actually a combination of models, rather than ensemble learning.

Reviewer #2: This paper studies the complementary performance of convolutional neural network and capsule neural network in segmentation of yeast cell microfluidic image classification, which has certain research value. However, this method seems to be limited in terms of novelty. The following are my main concerns:

1) The author compared three deep learning neural network methods. Due to the complementary performance, the whole composed of three most suitable single architecture models can achieve the highest overall accuracy, precision and recall rate. It is only a combination, so the technical novelty is low.

2) The motivation for combining the three models should be better explained.

3) A large number of experiments discussing the comparison results before and after data expansion, which is worthy of praise. But it is suggested that the advantages and disadvantages of different models can be discussed from other angles, such as loss and time.

4) We suggest you thoroughly copyedit your manuscript for language usage, spelling, and grammar. If you do not know anyone who can help you do this, you may wish to consider employing a professional scientific editing service.

6. PLOS authors have the option to publish the peer review history of their article (what does this mean?). If published, this will include your full peer review and any attached files.

Reviewer #1: No

Reviewer #2: No

---

## [Author Response · Author response to Decision Letter 0]

18 Dec 2020

Dear Editor,

We wish to submit the revised version of the manuscript entitled “Performance Comparison of Three Deep Learning Models on Classifying Microfluidic Images of Dividing Yeast Cells “ to be considered for publication in PLOS Computational Biology. 

This is our second submission (PONE-D-20-27932) of the manuscript since the previous submission (PONE-D-19-32184R1) was unfortunately rejected. In this revised version, we have modified the manuscript based on reviewers’ comments. This document has two parts as following order : 

- Response to “ second submission “ (reviewers' comments)

 - Response to “ first submission “ (reviewers' comments)

Each part contains individual responses to 1st reviewer’s comments and individual responses to 2nd reviewer’s comments. 

If you need further information, please do not hesitate to contact us.

Mehran Ghafari and Hong Qin

Department of Computer Science & Engineering,

SimCenter, University of Tennessee at Chattanooga

Email: ryg668@mocs.utc.edu

Response to second submission (reviewers' comments)

Reviewer #1 : 

1.The CapsNet (2017) is not the latest and the most effective. This domain develops fast and the authors should keep an eye on the literature.

Response: Line 48 

- The success of CapsNet lies in its ability to preserve additional information from input images by utilizing convolutional strides and dynamic routing instead of a max pooling layer. It has been argued that the spatial information of data has not been utilized in CNN models, including in the pooling function used to connect convolutional layers. For example, max pooling layers take only the most prominent values (e.g., pixels) from a previous convolutional kernel as input to the next layer. This issue considerably increases model inefficiency. In other words, CapsNet uses additional features of the dataset (e.g, spatial information) to improve the accuracy of a small dataset. These features are valuable especially in a medical dataset where there is a data limitation (e.g., images).

2. The methods of other people are not sufficient as highlights of study.

Response: Line 23 

- Deep learning is a sub-field of machine learning that has been applied in a wide range of applications [7, 8], and its developments are mostly driven by computational capacity and the accessibility of datasets [9]. In recent years, deep learning has increased in efficacy for image classification and is now a popular method for parsing image information [11]. Many innovations have been driven by creating models that perform well on benchmark datasets such as MNIST [12] (60,000 handwritten digits for training in a 28x28-dimensional vector space), CIFAR10 [13] (60,000 commonly used images in a 32x32-dimensional vector space), CIFAR100 [14] (500 training images grouped into 100 classes), ImageNet [15] (over 15M high-resolution images in over 22,000 classes), etc. The basic idea of deep learning is to create or learn a function that can map a high-dimensional input space into an output vector. For example, a high dimensional image can be filtered through neuron layers aiming for image classification and segmentation.

3. It is better that the authors can excavate certain medical meaning of this study.

Response: beginning in Line 1, we add the following 

- The budding yeast Saccharomyces cerevisiae is an effective model for studying cellular 

aging [1, 2]. The replicative lifespan of a yeast mother cell is defined as the total number 

of cell divisions accomplished or the number of daughter cells produced throughout its 

lifetime. Microfluidics is a fast-developing technology for the single-cell monitoring and 

imaging required in this context. In particular, microfluidic devices are partially automatic method to monitor cells development and classify cells which can speed up the manual process of cells lifespan estimation [3]. Typically, microfluidic images have relatively low resolution compared to confocal microscopic images that are often of high resolution [5], rendering unique challenges for microfluidics image processing [4]. For instance, microfluidic device materials, device coating, device volume, and area limitations increase capturing errors such as blurring, shifting focus, and trap deformation. Capturing the full progression of cellular replicative lifespans requires identifying both mother cells and daughter cells in full cell cycles [6]. Low image resolution hinders the automation of this process, demanding time-consuming, manual classifications of yeast replicative lifespans. Machine learning-specifically deep learning-could simplify this process.

4. The ensemble method proposed by the authors, is actually a combination of models, rather than ensemble learning.

Response: beginning in Line 313, we add the following 

- In machine learning, minimizing bias and variance errors is a challenging task. The 

weighted average ensemble model is one of the methods to overcome this issue that relies on two properties in machine learning [39]: creating an ensemble model such that the bias can be decreased at expense of increased variance, and creating an ensemble model such that the variance can be decreased at no expense to bias [40]. In general, there are two simple methods to combine several machine learning models and create an ensemble model with better performance. First, train a model (e.g., classifier) over multiple subsets of the training dataset, which leads to different models. Then, the individual model can have a prediction on the test dataset and the results can be averaged as an ensemble model. This method is useful when there is no other model available. The other method is to train various models on the same dataset and average the results on the test dataset. An ensemble model attains a synergistic betterment in overall performance including reproducibility and stability.

- Addition reference added ([40,41])

Reviewer #2: 

1. The author compared three deep learning neural network methods. Due to the complementary performance, the whole composed of three most suitable single architecture models can achieve the highest overall accuracy, precision and recall rate. It is only a combination, so the technical novelty is low.

Response: Start from Line 62 we add the following:

- The purpose of the current work is to compare deep-learning classification models of microfluidic images of dividing yeast cells. We compare three deep-learning neural network approaches to classify microfluidic trap images into 4 biological categories. This comparative study focuses on the performance of three models: two convolutional neural networks and a capsule neural network. The two convolutional neural networks contains 2 and 13 convolutional layers respectively. We also investigated ensemble models built from these three models. Due to dataset limitations, we investigated the effect of data augmentation on all three models.

2. The motivation for combining the three models should be better explained.

Response: Start in Line 313, we add the following: 

- In machine learning, minimizing bias and variance errors is a challenging task. The 

weighted average ensemble model is one of the methods to overcome this issue that relies on two properties in machine learning [39]: creating an ensemble model such that the bias can be decreased at expense of increased variance, and creating an ensemble model such that the variance can be decreased at no expense to bias [40]. In general, there are two simple methods to combine several machine learning models and create an ensemble model with better performance. First, train a model (e.g., classifier) over multiple subsets of the training dataset, which leads to different models. Then, the individual model can have a prediction on the test dataset and the results can be averaged as an ensemble model. This method is useful when there is no other model available. The other method is to train various models on the same dataset and average the results on the test dataset. An ensemble model attains a synergistic betterment in overall performance including reproducibility and stability.

- Addition reference added ([40,41])

3. A large number of experiments discussing the comparison results before and after data expansion, which is worthy of praise. But it is suggested that the advantages and disadvantages of different models can be discussed from other angles, such as loss and time.

Response: Start in Line 182, we added: 

- The advantages and disadvantages of individual model are mainly covered and explained in results and discussion part. Addition information can also be found from S2 Fig. 

4. We suggest you thoroughly copyedit your manuscript for language usage, spelling, and grammar. If you do not know anyone who can help you do this, you may wish to consider employing a professional scientific editing service.

Response: 

- We have revised the manuscript accordingly with a professional editor. 

Response to first submission (reviewers' comments)

This part contains the responses to the reviewer’s comments (see PONE-D-19-32184_reviewers_comments..pdf file in attachments). Each section has a line number corresponding to the reviewer’s concerns highlighted in yellow, the reviewer’s comments are in light-blue, and the author’s responses are in black.

Reviewer #1: 

Abstract: 

convolutional neural networks

Comment: Please give reference

Response: Ref added in introduction 

The capsule networks had the most robust performance at detecting one specific category of cell images.

-Kommt so etwas in den Abstract rein?

Ein Teil der Ergebnisse

-Google translation

Comment: Does something like that come into the abstract?

Response: Part of the results

In addition, extending classification classes and augmentation of the training dataset can improve the predictions of the biological categories in our study.

-Hat die Aussage einen Mehrwert?

Diese Aussage bezieht sich darauf, dass wir eine biologische Klasse 'mdC' in 'mddC' und 'mduC' aufteilen und die Trainingsdaten ergänzen.

-Google translation

Comment: Does the statement have any added value?

Response: This statement refers to the fact that we divide a biological class 'mdC' into 'mddC' and 'mduC' and supplement the training data.

Introduction to conclusion: 

Line 7: Typically,

Comment: Please give a reference to other papers which use time intervals with low resolution.

Response: Ref added 

Line 9: unique challenges

Comment: Please list some examples of unique challenges.

Response: For instance, microfluidic device materials, device coating, device volume and area limitation increase capturing errors such as blurring, shifting focus, trap deformation.

Line 13: [5]

Comment: Please explain how this reference fits to the statement that "full automation of this process is often hindered by low image resolution"

Response: Ref [5] moved to previous sentence

Line 15: wide range of applications,

Comment: Please add references and specify some applications with connections to this topic.

Response: two references are added

Line 16: The development of deep learning is driven by its ability to understand and infer information from data such as speech, text, and images [6].

Comment: Why a reference to "Learning Deep Structure-Preserving Image-Text Embeddings" ?

Response: The highlighted sentence removed 

Line 23: MNIST

Comment: Please reference to MNIST

Response: Ref added 

Line 24: CIFAR10

Comment: Please add a reference to CIFAR10

Response: Ref added 

Line 25: CIFAR100

Comment: Please add a reference to CIFAR100

Response: Ref added 

Line 26: The basic idea of deep learning is to create or “learn" a function that can map a high-dimensional input space into an output vector.

Comment: Please explain it in more detail.

Response: For example, a high dimensional image can be filterized through neuron layers aiming for image classification and segmentation.

Line 29: In classification,

Was ist das Hauptaugenmerk in diesem Absatz?

-Klassifikation oder Arten der CNN?

-Oder Probleme bei CNN?

Der Absatz konzentrierte sich hauptsächlich auf die Bildklassifizierung mit CNN (entsprechend modifiziert).

-Google translation

Comment: What is the main focus in this paragraph?

Comment: Classification or types of CNN?

Comment: Or problems with CNN?

Response: Sales mainly focused on image classification with CNN (modified accordingly).

Line 31: In image classification problems, the convolutional neural network (CNN) is the primary type of deep learning model 

employed.

Comment: Give some references to your statement

Response: Ref added 

Line 38: CapsNet,

Comment: Please give a reference to the traditional CapsNet architecture.

Response: Ref added 

Line 52: trained each model with consideration of the effect of data augmentation. Finally, we showed that an ensemble of the top three models performs better than using each individual model alone, leading to a good \\collaboration" among these models. In addition, data augmentation and splitting a class into two classes could be an effective approach for some models based on the type of dataset and model architecture.

Comment: No comments 

Response: Modified accordingly 

Line 62: in S1 Table of the

Comment: Supporting Table 1 is not self-explanatory/clearly structured.

Comment: It is not clear which combinations had been made.

Comment: Why write three times colums with the same options for CNN-2/CNN-13a dnd CapsNet?

Comment: This information is redundant to some extend. Please structure it more clearly.

Response: Grid search and augmentation options table. The grid search option table used for all models and augmentation features applied when the augmentation in grid search option set to "True.

Fig 1: pixels.

Comment: Mention that there lie 10 minutes between two consecutive measurements. The caption and the picture should be self-explanatory.

Response: Modified accordingly

Line 85: However, for easier understanding from a biological point of view, mddC and mduC classes are merged and labeled mdC after the testing process.

Comment: In how many cases does this occur?

Comment: [Am I going right with my assumption that it's 2%?]. If a significantly higher Comment: proportion of the data set is affected, a more detailed description of the data set used would be necessary.

Comment: Changed with more explanatory details 

Response: Modified with more explanatory details 

Line 90: the 4 biological classes,

Comment: If no additional weighting is apllied here, to generate "the 4 biological classes" this labeling should not be used, as this is only confusing and has no added value for the classification carried out.

Response: Modified with more explanatory details 

Line 99: A 2x2 kernel size used for max-pooling and 25% dropout applied for the second 99 layer as the model architecture is shown in Fig 3 (a).

Comment: Please add some justifications for your modifications of your architecture.

Response: Recommended by default and there was not any modification 

Line 145: https://github.com/QinLab/GhafariClark2019.

Comment: A more precise commenting as well as other folder directories would be desirable.

Comment: It was also not possible to understand how the evaluations were carried out.

Comment: A small introductory text in the code would be desirable. 

Comment: After a few attempts the code review was stopped.

Response: The code requires GPU (cuda 9.2) and python libraries (e.g. openCV) 

Line 166: with pre-trained weights

Comment: Please explain which weights were taken, more specifically how you found the weights you used for this analysis.

Response: Explained in more details with reference

Line 169: are similarities

Comment: Similarities in which way? Please explain.

Response: Modified accordingly (due to pattern and orientation) 

Line 171: cell. Based on this observation, we split all images with two cells into two separate classes; in the first class, the daughter cells are on top of mother cells (upward-oriented, mduC class), and in the second class the daughter cells are below the mother cells (downward-oriented, mddC class), as illustrated in Fig 2 (c).

Comment: Between line 77 and 91 the reader is told that first the 5 biological classes were introduced and then these were combined to 4 classes. Here again it is argued that similarities between exC and mdC were considered which is why mdC was divided into mddC and mduC.

Comment: What was there first? Labels with the 5 classes or labels with the 4 classes?

Comment: Please structure the paper in a chronologically correct order of class construction.

Response: Modified accordingly: In many cases, a single mother cell appears as two cells (due to dynamic shape and low image resolution) and the daughter cell is above the mother cell.

Line 179: dataset. However, the results for mddC and mduC classes are averaged

Comment: What do you mean with averaged? 

Comment: Do you talk about renaming?

Comment: if (result_label in c("mddC","mduC")){ New_label="mdC")

Comment: if (result_label not in c("mddC","mduC")){ New_label=result_label)

Response: Modified accordingly: Merged (not averaged)

Line 188: was improved to 92%. Moreover,

Comment: Please mention how much it was before the augmentation. 

Response: Modified accordingly (from 87% to 92%)

Fig 4: Fig 4. Comparison results for classification models.

Comment: Representation of picture 4 is not intuitive. The results could be displayed in tabular form. This would allow a more accurate comparison than using bar charts. In addition, the width of the bar charts is different. All this leads to situations where e.g. the precision values of mdC (%) do not show how they change.

Response: Modified accordingly 

Line 254: Each deep learning model has its own profiles of misclassifications 

Comment: The Average Results of Figure 4 do not address the different number of cases within the individual classes. Likewise, Accuracy's values are not shown separately by class. Statements made from Figure 4 and Figure 5 are not transparent.

Response: Modified accordingly 

Fig 5: Fig 5. Each of the three deep learning models has idiosyncratic error profiles

Comment: There is no explicit connection between the columns Before augmentation & After augmentation and the very right column. The reader cannot understand how much of "correct pred bf augmment" can be traced back to which class in which proportion.

Response: Modified accordingly 

Line 281: straightforward ensemble method

Comment: Pleas eexplain in mord detail what/how you have done it.

Response: More details are added 

S1_Table: Grid search and augmentation options.

Ich finde es schwer nachzuvollziehen wie der Grind-Search durchgeführt wurde. Könntet ihr mir eine genauere Beschreibung hierfür geben?

Entsprechend geändert 

-Google translation

Comment: I find it difficult to understand how the grind search was carried out. Could you give me a more detailed description of this?

Response: Changed accordingly

S2_table: The results of precision, recall and accuracy for all models.

Comment: Please specify additionaly the accuracy seperated by their class. In addition add the sample size of each class used for classification and the sample size used for testing.

Example: Table:

Also specify the values for all tables separately for mduC and mddC.

 Training images Testing Images Accuracy

CNN-2 mC 111 27 XY%

 exC 112 42 XY%

 mdC 113 43 XY%

 mddC 114 45 XY%

 mduC 115 49 XY%

Overall Accuracy 

Response: S4_Table_accuracy added

Reviewer #2: 

Line 26: ImageNet

Comment: Please provide a citation for this specific dataset.

Response: Ref added 

Line 9: filterized 

Comment: Please reformulate accordingly.

Response: Changed accordingly 

Line 33: The output is a vector that the size of the output vector depends on the number of classes

Comment: Please reformulate this specific sentence accordingly!

Response: which the output vector depends on the number of classes.

Line 35: because they are mainly designed for 2-dimensional (or higher) input

tensors 

Comment: Is this really a justification for the successful use of CNNs in the domain of image classification? What about the hierarchical construct characterizing CNNs that enables such a model to slowly but successfully learn relevant representations adapted to the specific task at hand? Please reformulate this specific sentence by adding a better and pertinent justification.

Response: The CNN-2 and CNN-13 are used for comparison purposes considering the effect of number of layers in the model. 

Line 43: in datasets

Comment: involving small sized datasets." Please correct accordingly

Response: involving small sized datasets

Line 23: A recent study showed that CapsNet could classify fluorescent microscopic images 

Comment: At which extent? Please be more specific.

Response: For example, max pooling layers take the most prominent values (e.g. pixels) from a previous convolutional kernel as input to the next layer. A recent study showed that CapsNet could classify fluorescent microscopic images [38]. The model illustrated improvement in accuracy on datasets such as MNIST, yet it is computationally expensive as training time increases substantially. In [19], authors claimed that the CapsNet can achieve near state-of-art performance on the MNIST dataset using 10 % of whole dataset. 

Line 54: of the top three models

Comment: How many models have been assessed? If there are just three models, please correct the phrase accordingly.

Response: We showed that an ensemble of the top three models performs better

Line 56: could be an effective approach for some models based on the type of dataset and model architecture

Comment: An effective approach to achieve what exactly? Please be specific.

Response: In addition, dataset augmentation and splitting a class into two classes could be an effective approach for some models based on the type of dataset and model architecture. 

Line 64: S1 Table

Comment: Missing table 

Response: added accordingly 

Line 79: the 5 categories

Comment: The authors mean "... the following 5 categories ...". Please correct accordingly.

Response: We trained the deep learning methods using the 5 categories based on cell numbers and their relative positions: a trap with no cell (nC), a trap with a single mother cell (mC), a trap with mother and one upward-oriented daughter cells (mduC), a trap with mother and one downward-oriented daughter cells (mddC), and a trap with more than two cells (exC).

Line 86: but

Comment: Please delete!

Response: removed accordingly 

Line 95: it is also termed

Comment: " ... it is also referred to as ... " Please correct accordingly.

Response: it is also referred to as the baseline CNN architecture.

Line 97: the,

Comment: Please delete!

Response: removed accordingly 

Line 98: The stride for the second layer is 2

Comment: What about the stride of the first layer? 

Response: The stride for the first and second layers are 1 and 2 respectively. Added to in line 103 and explained in figure 4. 

Line 108: HasanPour et al. [28]. HasanPour et al. [28]

Comment: Please correct accordingly (successive repetition of the citation).

Response: Modified accordingly 

Line 114: CNN-13 has 2-25 times fewer parameters than the popular models.

Comment: Which models are the authors referring to? Please be specific

Response: Modified with more explanatory details 

Line 117: In addition, batch normalization and 25% dropout applied to all layer

Comment: Please correct accordingly.

Response: removed accordingly 

Line 124: Instead of generating scalar output as used in CNNs, a capsule layer in CapsNet generates a vector as output from convolutional kernel inputs, where the length of the vector represents how likely it is that a feature from the previous layer is present, and the values of the vector are an encoding of all the affine transformation of the kernel inputs

Comment: This particular description does not help the reader to understand this specific aspect of capsule networks. Please improve the description and use equations as well as useful depictions where needed

Response: Modified with more explanatory details and equations

Line 143: width, and height

Comment: Do you mean scaling the image? Please correct accordingly.

Response: no scaling, removed accordingly 

Line 145: We used 4,104 trap images for training and 896 for testing. We augmented the training images, which resulted in 99,380 training images

Comment: How about the validation set? Which proportion of the training material was used as validation set in order to perform the selection of the models used in the ensemble as well as the performed grid search? If the grid search as well as the selection of the models were performed on the test set, the resulting ensemble is an overfitting model and wont be able to generalize well. If this was the case, the authors have to repeat all the conducted experiments, by defining a validation set used for the grid search and for the models selection. Once this is done, the models have to be trained and finally tested on the test set.

Response: Explained in more details 

Line 167: In general, the transfer learning is a neural network that starts with pre-trained weights which models can learn weights in a shorter time.

Comment: Please reformulate this specific sentence. It does not make much sense.

Response: removed accordingly 

Line 169: The other concept is splitting classes and use pre-trained weight from [28] which the method has some similarity to transfer learning as both methods attempt to make it easy for the models to learn weights; however, these approaches come from different angles.

Comment: Please reformulate and specify the message more clearly

Response: (Modified accordingly) For instance, we notice that there are size, pattern and orientation similarities between the exC class and the mdC cell class. In many cases, a single mother cell appears as two cells (due to dynamic shape and low image resolution) and the daughter cell is above the mother cell.

Line 179: At the highest level, creating mddC and mduC classes improved the homogeneity of the two classes and helped the situation where the neural networks were able to more easily learn the differences of the mduC class and the exC class without having to learn that the mddC and mduC class are the same. It is important to notice that all training and testing activities are based on the computed 5 classes dataset. However, the results for mddC and mduC classes are merged and labeled as mdC for easier biological understanding as shown in Fig 2 (b).

Comment: So, the models are trained based on the five-class problem, and also tested based on the five-class problem. But the results specific to both classes mddC and mduC are subsequently aggregated into a unique class mdC. How is this aggregation performed? How about the data distribution concerning both mddC and mduC classes?

Response: explained accordingly including figure 2 modification 

Line 189: Fig 4

Comment: Are these results depicting the classification performance on the test set uniquely? Or are also some reclassification results (on the training set) depicted?

Response: Modified figure and additional explanation 

Line 191: Fig 5

Comment: Same issue or question as in Fig. 4.

Response: Modified figure and additional explanation

Line 191: augmentation

Comment: Please refer to this as data augmentation!

Response: modified accordingly 

Line 192: the accuracy of prediction

Comment: Recall of the mdC class?

Response: refer to Figure 5 and 6 

Line 194: 92%

Comment: Inconsistent results: 809/896 = 90.29% ?

Response: corrected accordingly 

Line 206: Augmentation of training data also led to more stable CNN-13 models as seen when changes of the cost functions during training became more smooth with augmented datasets

Comment: Is there a specific plot that shows this specific aspect of data augmentation for the model CNN-13? Please provide such a comparison plot (with and without data augmentation).

Response: refer to S2 Fig. 

Line 233: S1 Table

Comment: Missing table 

Response: added accordingly 

Line 233: We picked the best-performing CapsNet model for this study

Comment: Steel how was the grid search performed? Was the grid search performed using the test set as validation or a specific validation set? The grid search should be performed on a validation set since the test set should not be seen during the optimization of the model. If the parameter optimization step was done using the test set, all the depicted experiments should be performed at new using a validation set which do not include any of the samples belonging to the test set. And the proportion of data used as validation set as well as the selection process of the samples should be described thoroughly.

Response: refer to S1 Table 

Line 236: In Zhang et al. [37], a close range of accuracy was reported for fluorescent images.

Comment: What does this mean exactly? Please be more specific.

Response: removed accordingly 

Fig 4: Results without augmentation

Comment: Please refer to each of the depicted plots with Fig. 4.1, Fig. 4.2, Fig 4.3, ...

Response: modified accordingly ( see fig 5) 

Fig 4 : Accuracy

Comment: Results inconsistency: if one takes a look at the bar plots of Test Results (all classes), CNN-13 depicts the highest overall accuracy, followed by CNN-2 and CapsNet. But in this depiction, the data is showing something complete different. What are these results referring to? Please correct accordingly.

Response: corrected accordingly ( see fig 5)

Fig 4 (legend): The table is presenting the correct and misprediction results of mddC and mduC classes without augmentation for all three models.

Comment: The current depiction of the results is confusing: how many samples belong to the class mdc? normally we should have total number of mdc samples = correct prediction + miss prediction. And this specific number should not vary from one model to another since it is the same classification task. But when we look at the number (correct prediction + miss prediction) for each model: cnn-2: 380 != cnn-13: 371 != 495 ?

Response: corrected accordingly ( see fig 5)

Fig 4 (legend): In the bar graphs, the precision an recall are shown for individual class based on biological interpretation classes (four). The mean and total tested images results are presented for all models

Comment: These results seem to be inconsistent: e.g. CNN-2: accuracy = 780/896 = 87.05% but looking at the Average results (all classes) bar plot, CNN-2: accuracy > 90%?

Response: corrected accordingly ( see fig 5)

Line 250: S4 Table

Comment: This table appears after the S5 Table but is named S4 Table. Please correct accordingly! Moreover, what does count stand for? Is it the total number of samples in each class? Please use a better and more specific column name.

Response: corrected accordingly

Line 251: preformed 

Comment: "performed" ... Please correct accordingly.

Response: corrected accordingly

Line 250: the most of predictions

Comment: " ... most predictions ..." ... Please correct accordingly.

Response: corrected accordingly

Line 253: By contrast,

Comment: "In contrast, ..." ... please correct accordingly.

Response: corrected accordingly

Line 255: even though it has a skeleton architecture

Comment: What is a skeleton architecture? Please provide more information and be more specific..

Response: removed accordingly

Line 259: With the additional 11 more layers

Comment: "With the additional 11 layers ..." ... Please correct accordingly.

Response: corrected accordingly

Line 260: a moderate 6% increase from CNN-2.

Comment: What do the authors point out with moderate? Please perform significance tests!

Response: With the additional 11 layers and much more training time, CNN-13 improved the overall accuracy to 98%, a partial 6% increase from CNN-2.

Line 261: that performance of CNN-13 is not substantially changed

Comment: "... that the performance ..."

Response: corrected accordingly

Line 263: which is around 16 times lower than CNN-2

Comment: What are the authors referencing to? The comparison of both performances (CNN-2 or CNN-13 with data augmentation)? Please correct accordingly.

Response: explained accordingly (performance of CNN-13 is not substantially changed by applying dataset augmentation. Fig 6 shows that dataset augmentation improved the total prediction of CNN-13 by 0.22%, which is around 16 times lower than CNN-2 and decreased the total misprediction by 8.3%, which is considerably lower than CNN-2)

Line 277: We also investigated misclassification behavior of individual models for the mC, mdC, and exC classes (see Fig 4 and Fig 5). In terms of correct-prediction balance between mddC and mduC, Fig 4 demonstrates that all the models had relatively close range of prediction for mddC and mduC (before and after augmentation). In terms of

misprediction, the CNN-2 model had opposite behavior of the CNN-13 and CapsNet

models. For CNN-2, the mduC class had a higher percentage of misclassification for the

mC class, and the exC class had higher misclassification for the mddC class. For CNN-13 and CapsNet, the mddC class had a higher percentage of misclassification for the mC class, and the exC class had a higher misclassification for mduC. These comparisons indicate that why we consider an ensemble model as an alternative.

Comment: Please use the annotation Fig 4.1, Fig 5.1 ... for the sake of clarity. Moreover, the depicted results are inconsistent and the used depiction is kind of misleading. Please correct accordingly.

Response: corrected accordingly( see fig 5 and fig 6)

Line 288: Inasmuch each

Comment: "Inasmuch as each ..." ... please correct accordingly.

Response: corrected accordingly

Line 289: models archiving better performance

Comment: please correct accordingly.

Response: corrected accordingly

Line 291: (see S2 Fig).

Comment: Please use the same nomenclature CNN-2, CNN-13 and CapsNet in this figure, for the sake of clarity. Moreover, please use Fig S2.1, Fig S2.2, ... for each of the depicted architecture..

Response: corrected accordingly

Line 291: The outcome results from all three models indicate that

weighing the predictions by overall model accuracy achieves slightly better

performance [38]

Comment: Have the authors tested each of the depicted architectures in order to come to this conclusion? Such an experiment should be undertaken.

Response: explained in more details

Line 293: Thus, models in the ensembles presented are weighted by their overall

test set accuracy and misclassifications.

Comment: This is wrong! The weights should be optimized on a validation set and subsequently applied on the test set. The test set should not be seen during the optimization of the parameters of the architecture. These experiments have to be computed as new.

Response: explained accordingly

Line 296: Therefore, we found that the three-member ensembles, No.4 (see S2 Fig)

outperformed all of the two-member ensembles.

Comment: Where are the results of these experiments?

Response: explained accordingly (since the result of ensemble 1 to 3 were almost similar to individual model, we only represented result of ensemble No. 4) 

Line 301: In terms of ensemble No.4 misclassifications, each of the ensemble models are obviously misclassifications of at least one of the three models.

Comment: Please correct this sentence accordingly.

Response: modified accordingly 

Line 306: the image pre-processing differently from data augmentation.

Comment: What are the authors pointing at? Please reformulate!

Response: explained accordingly (there are still improvements to be made in image pre-processing ( e.g. image resolution). 

Fig 5: Results with augmentation

Comment: Please refer to each of the depicted plots with Fig. 5.1, Fig. 5.2, Fig 5.3, ...

Response: modified accordingly ( see fig 6) 

Fig 5 : Test Results ( all classes)

Comment: The accuracy results of the Ensemble are missing! Please correct accordingly.

Response: see fig 6 yellow bar 

Fig 5 : Accuracy

Comment: These results are completely inconsistent with the ones depicted in the bar plots of Test Results (all classes). Please correct accordingly!

Response: each results labeled accordingly ( see fig 6) 

Fig 5 (legend): The mispredictions results for CNN-13 and CapsNet were one of motivation to generate ensemble models.

Comment: Please reformulate and correct this sentence accordingly!

Response: corrected accordingly ( see fig 6)

Fig 5 (legend): In the bar graphs, the precision an recall are shown noticeable improvement for all models after augmentation

Comment: What does noticeable refer to? Do the authors mean significant? Please perform and describe the significance tests performed!Response: corrected accordingly ( see fig 6)

Line 313: Microfluidiic

Comment: Please correct accordingly!

Response: corrected accordingly

---

## [Decision Letter · Decision Letter 1]

1 Feb 2021

Complementary Performances of Convolutional and Capsule Neural Networks on Classifying Microfluidic Images of Dividing Yeast Cells

PONE-D-20-27932R1

Dear Dr. Ghafari,

We’re pleased to inform you that your manuscript has been judged scientifically suitable for publication and will be formally accepted for publication once it meets all outstanding technical requirements.

Kind regards,

Friedhelm Schwenker

Academic Editor

PLOS ONE

Additional Editor Comments (optional):

Reviewers' comments:

Reviewer's Responses to Questions

**Comments to the Author**

1. If the authors have adequately addressed your comments raised in a previous round of review and you feel that this manuscript is now acceptable for publication, you may indicate that here to bypass the “Comments to the Author” section, enter your conflict of interest statement in the “Confidential to Editor” section, and submit your "Accept" recommendation.

Reviewer #1: (No Response)

Reviewer #2: All comments have been addressed

2. Is the manuscript technically sound, and do the data support the conclusions?

Reviewer #1: Partly

Reviewer #2: Yes

3. Has the statistical analysis been performed appropriately and rigorously? 

Reviewer #1: Yes

Reviewer #2: Yes

4. Have the authors made all data underlying the findings in their manuscript fully available?

Reviewer #1: Yes

Reviewer #2: Yes

5. Is the manuscript presented in an intelligible fashion and written in standard English?

Reviewer #1: Yes

Reviewer #2: Yes

6. Review Comments to the Author

Reviewer #1: I am sorry to refuse it for your further improvement. There is a lack of innovation in methods or providing new data to other researchers. After this revision, I still cannot see the expected significant contribution of this study. Furthermore, the performance of their method is not convincing.

Reviewer #2: accept. The author has answered all required questions, and all responses conform to the format specification.

7. PLOS authors have the option to publish the peer review history of their article (what does this mean?). If published, this will include your full peer review and any attached files.

Reviewer #1: No

Reviewer #2: No

---

## [Editor Report · Acceptance letter]

18 Feb 2021

PONE-D-20-27932R1 

Complementary Performances of Convolutional and Capsule Neural Networks on Classifying Microfluidic Images of Dividing Yeast Cells 

Dear Dr. Ghafari:

I'm pleased to inform you that your manuscript has been deemed suitable for publication in PLOS ONE. Congratulations! Your manuscript is now with our production department. 

Kind regards, 

on behalf of

Dr. Friedhelm Schwenker 

Academic Editor

PLOS ONE